# Effect of Microbiological Growth Components for Bacteria-Based Self-Healing on the Properties of Cement Mortar

**DOI:** 10.3390/ma12081303

**Published:** 2019-04-20

**Authors:** Xin Chen, Jie Yuan, Mohamed Alazhari

**Affiliations:** 1State Key Laboratory of Safety and Health for In-Service Long Span Bridges, JSTI Group, Nanjing 210019, China; xin.chen@alu.hit.edu.cn; 2School of Transportation Science and Technology, Harbin Institute of Technology, Harbin 150090, China; 3Department of Civil Engineering, University of Tripoli, Tripoli 13110, Libya; s.elazhari@uot.edu.ly

**Keywords:** cement, concrete, bacteria, self-healing, hydration, strength, microstructure

## Abstract

Previous studies of bacteria-based self-healing concrete have shown that it is necessary to encapsulate and separate the self-healing ingredients (bacteria, nutrients, and precursors) in the concrete so that when a crack forms, capsules rupture, which allows the self-healing ingredients to come together and precipitate calcite into the crack. Because of the shearing action in the concrete mixer, there is a chance that these capsules, or other carriers, may rupture and release the self-healing ingredients. This would affect the efficiency of self-healing, but may detrimentally affect the concrete’s properties. This work investigated the effects of multi-component growth media, containing germination and sporulation aids for the bacterial aerobic oxidation pathway, on the basic properties of fresh and hardened concrete instead of the potential self-healing efficiency in a structural service. Tests were carried out to measure the effects of growth media on air content, fluidity, capillary absorption, strength development of cement mortar following corresponding standards, hydration kinetics, setting properties, and the microstructure of cement paste, according to certain specifications or using specific machines. The research has demonstrated that a multi-constituent growth media will not have a significant effect on the properties of concrete in the proportions likely to be released during mixing. This important conclusion will allow further development of these novel materials by removing one of the key technical barriers to increased adoption.

## 1. Introduction

Concrete is one of the most popular construction materials used worldwide because of its high compressive strength, relatively low cost, and the wide availability of its component materials. For many decades, researchers have been attempting to improve the durability of concrete to better enable structures to resist early age cracking, freeze/thaw cycles, chemical attack, corrosion, and other environmental factors. Durability concerns are often related to cracking, which, in the tensile or cover areas, will lead to rapid degradation. The quantity of cracks increases when the quality control and quality assurance of the concrete production are poor [1]. In Europe, 50% of the annual construction budget is spent on repair work and the cost for work on reinforced concrete increases when the cracks are not easily visible or accessible. A potentially beneficial method to reduce the maintenance cost of concrete cracks and improve durability is the use of self-healing concrete.

A much-researched form of self-healing is that of microbiologically-induced calcite precipitation (MICP) in which the metabolic activity of bacteria convert calcium precursors into calcium carbonate. Calcite as the most stable polymorph is generally precipitated [2]. 

One means of achieving MICP in concrete is via enzymatic hydrolysis of urea. In this case, the bacteria convert urea into ammonia and carbon dioxide. This reaction increases the pH from neutral to alkaline conditions, which results in the formation of carbonate [3]. Although the enzymatic hydrolysis of urea has proven to provide effective self-healing in concrete, it has been criticized because the formation of ammonia is environmentally damaging [4]. Moreover, urea has limited long-term stability in the alkaline environment present in concrete. Therefore, this technique is generally considered best suited as an externally applied repair treatment rather than as a self-healing agent [5,6].

Consequently, other research has focused on an alternative pathway of aerobic oxidation [7]. In this pathway, the healing requires the bacteria to act as a catalyst for the conversion of an organic calcium salt (precursor), typically calcium acetate (CaC_4_H_6_O_4_) or calcium lactate (CaC_6_H_10_O_6_), to CaCO_3_ in the presence of water, oxygen, and nutrients. As shown in Equations (1) and (2):CaC_4_ H_6_ O_4_+4O_2_→CaCO_3_+3CO_2_+3H_2_O(1)
CaC_6_H_10_O_6_+6O_2_→CaCO_3_+5CO_2_+5H_2_O(2)

The only by-products of the conversion of these calcium salts to calcite are carbon dioxide and water, which are compatible with concrete. Furthermore, a weak carbonic acid may form that will lead to carbonation of calcium hydroxide within the concrete, which potentially leads to enhanced autogenous healing. 

A range of spore-forming alkaliphilic bacteria have been used for this pathway, including *Bacillus cohnii*, *Bacillus pseudofirmus* [8], and *Bacillus alkalinitriculus* [9]. The bacteria are normally added in the form of spores in order to be able to resist the harsh environment inside the concrete [10]. The appropriate choice of bacterial spores is dependent on the environmental conditions in which the concrete will be used.

To permit the germination, growth, and multiplication of bacteria within the concrete, nutrients are needed, typically consisting of carbon and nitrogen sources. Together with the organic calcium salt, these nutrient sources can be considered as the growth media (GM). Generally, standard biological growth media can be used such as B4 or modified B4 medium consisting of approximately 0.4–0.5% yeast extract, 0.5% glucose or dextrose, and 0.25–1.5% calcium acetate. However, yeast extract as the sole nutrient source, in combination with a calcium salt, is most often used [9]. In general, growth media are used at approximately 4–5% by the mass of the cement [9,11]. The healing process, which may take up to 100 days for a crack of around 0.5 mm in width, may be accelerated by the addition of a buffering compound, e.g. sodium silicate [12]. However, it has been suggested that the germination and growth of bacteria in concrete and precipitation of calcite is improved by the use of multi-constituent self-healing agents comprising microbiological growth components [11].

State-of-the-art research has attempted to increase the functionality of self-healing concrete by incorporating the self-healing agents in various carriers, including macro-encapsulation, microencapsulation, and vascular systems [13]. One important consideration in carrier selection is the survivability during concrete mixing and placing. Only when the majority of the carriers remain intact after mixing and placing, can the cargo inside potentially contribute to the self-healing of in-service structures. The cementitious hollow tube is one typical macro-encapsulation technique that was designed with good survivability during the real concrete mixing process [14,15]. However, it is not suitable for bacteria-based self-healing concrete due to its size and alkalinity. Microencapsulation of spores is commonly used in bacteria-based self-healing concrete, but water-soluble growth components for bacteria are not easily encapsulated in this form [13,16]. Impregnation in porous materials such as lightweight aggregates and hydrogels is another commonly suggested protection method for the bacteria spores and the growth components [9,11,17]. However, concerns remain relating to the effect of the nutrients and precursors on the setting and hardening of concrete should they be released accidentally due to the failure of the encapsulate during mixing and placing. This can occur since the porous carriers always have a lower strength than either hydrated calcium silicate or traditional aggregates [18]. For example, Wiktor and Jonkers [12] suggested that calcium acetate delays the setting and hardening of concrete, while Bundur et al. [19] suggested that the hydration kinetics and compressive strength of concrete are significantly altered by the presence of yeast extract. Amiri and Bundur [20] utilized corn-steep liquor (CSL) instead of yeast extract as an alternative carbon source to avoid the adverse effect on concrete properties such as initial setting time and compressive strength. Consequently, the aim of this research was to investigate the potentially detrimental effects of accidentally released growth components on the basic properties of fresh and hardened concrete instead of the self-healing efficiency in structural service. Although growth components were added directly (both individually and as a multi-component addition) into fresh mortar or cement paste in the research, they were designed to simulate the accidentally released growth media from an encapsulated system instead of building up a non-encapsulated self-healing system. Properties of fresh concrete (air content and fluidity), capillary absorption, strength development, hydration kinetics, setting, and microstructure were covered in the study.

## 2. Materials and Methods

### 2.1. Mortar Tests

The growth media used in this research were based on initial microbiological studies and were selected to enable the germination of bacterial spores, growth of bacterial cells, precipitation of calcite, and sporulation of bacteria for the healing of concrete [8]. The functions of the individual components in the growth media are provided in Table 1. Overall, four types of growth media were tested. These included the multi-component growth mediaGM-1 in addition to the use of the three major components of the growth media (calcium acetate, sodium citrate, and yeast extract) added individually as controls (See Table 2).

Tests were carried out to measure the effect of growth media on the air content, fluidity, capillary absorption, strength development, hydration kinetics, setting, and microstructure of paste and mortars.

#### 2.1.1. Mix Proportions

For mortar tests, growth media was added at a concentration equal to 0.5% of the mass of cement, which reflects the accidental release of around 10% to 12% of all self-healing agents from failure of a capsule during mixing and placing of the concrete, which is consistent with what is identified in practice [21].

For tests using the individual major components (calcium acetate, sodium citrate, and yeast extract), these components were added at a dosage that was equivalent to that present when the growth media was used at 0.5% of the mass of cement (i.e. 0.28%, 0.09%, and 0.06% of the mass of cement, respectively). The quantities were considered sufficiently minor to not warrant a reduction in cement or sand content to maintain yield, as shown in Table 3. 

Mortars were made at a water/cement ratio of 0.5. The cement used was CEM II/B-V 32.5R (Lafarge, Paris, France)and conformed to BS EN 197-1 [22]. Mortars were prepared in accordance with BS EN 196-1 [23],using EN standard sand. Table 3 gives the mix proportions for each mortar.

#### 2.1.2. Test Methods

Mortars were tested for the following properties: surface tension, air content, fluidity, capillary absorption, and strength.

The surface tension of the simulated pore solution was measured using hollow glass capillary tubes [24]. The pore solution was simulated using solutions of the growth media alone rather than including the cement paste, which would complicate analysis. Solutions of the growth media were saturated because the pore fluid in hardened mortar or concrete is eventually saturated. This is because free water is bound in hydration products or evaporates during setting and hardening processes.

In the test, the absorbed solutions in the hollow glass capillary tubes rise to a certain height by the force of surface tension. According to Laplace’s Equation:2πrσcosθ = πr^2^ρgh,(3)
where,
r is the inner radius of the capillary tube;σ is the surface tension coefficient of the tested liquid;θ is the contact angle;ρ is the density of the tested liquid;g is the gravitational acceleration;h is the height of the liquid column in the capillary tube.

The surface tension of the tested solution is proportional to the product of the density and the height of the liquid column in the capillary tube. Therefore, this can be calculated by a comparison with pure water where the surface tension is 72.75 mN/m at 20 °C (Equation (4)):∆F_x_ = 72.75(ρ_x_ × H_x_)/(ρ_w_ × H_w_).(4)
where,
△F_x_ is the surface tension of the tested solution;ρ_x_ is the density of the tested solution;ρ_w_ is the density of pure water;H_x_ is the height of the tested solution in the capillary tube;H_w_ is the height of pure water in the capillary tube.

Air content tests were conducted in accordance with BS EN 1015-7 [25] using Method A (the pressure method). In the meter, there is a sealed air chamber and a mortar sample container, connected to each other through an air valve. When the air valve is open, the pressures of the two chambers are equalized. The reduced pressure of the air chamber reflects the air content of the tested mortar.

Mortar fluidity tests were conducted in accordance with BS EN 1015-3, 1999 [26]. The tested fresh mortar is demolded from a defined mold on the flow table and then subjected to a number of vertical impacts. The flow value is the mean diameter (mm) of the tested mortar after these vertical impacts.

Capillary absorption tests on mortars were conducted in accordance with BS EN 480-5, 2005 [27]. After 28 days curing in a conditioning room, samples were weighed and then contacted with 3 mm deep water. The measurement was taken and the calculation was done according to the following formula (Equation (5)).
C_A_ = (M_j_ − M_o_)/1600,(5)
where,
M_o_ is the mass (g) after curing for another 7 days;M_j_ is the mass (g) after the required absorption time.

Compressive strength tests and flexural strength tests were conducted in accordance with BS EN 196-1 [23] at 28 days. 

### 2.2. Paste Tests

Pastes, rather than mortars, were used to assess the effect of growth media on hydration kinetics, setting time, and for microscopy.

For the studies on hydration kinetics using cement pastes, the growth media were added at 0.1%, 0.3%, and 0.5% of the mass of cement. The individual major components (calcium acetate, sodium citrate, and yeast extract) were added at a dosage that was equivalent to what was present when the GM-1 was used at 0.5% of the mass of cement. Another multi-component growth media GM-2 in which sodium citrate is deprived, was also introduced in the tests. The water/cement ratio was 0.5 for all tests.

Setting time tests were conducted in accordance with BS EN 196-3:2005 [28]. The water content was 165 g per 500 g of cement. GM-1 was added at 0.5% of the mass of cement. The individual major components (calcium acetate, sodium citrate, and yeast extract) were added respectively at a dosage that was equivalent to that present when GM-1 was used at 0.5% of the mass of cement.

The times of the initial set and final set are the two test indicators in setting time tests. The initial set is a state in which tested paste begins to lose plasticity, while the final set is a state where the tested paste completely loses plasticity.

SEM scanning (FEI, Hillsboro, USA) was used to observe the microtopography of the cement paste. GM-1 was added at 0.5% of the mass of cement.

## 3. Results and Discussion

### 3.1. Surface Tension of Saturated Solutions

The effects of growth media on the surface tension of saturated solutions are shown in Table 4.

As shown, the GM-1 resulted in a decrease in surface tension, even though two of the major components, which are calcium acetate and sodium citrate, increased surface tension.

### 3.2. Air Content of Mortar

The effects of growth media on the air content of mortars are shown in Figure 1. Results were obtained from averages of triplicate samples. The control mortar had an air content of 2.1% of the volume, which is typical of mortars. It was observed that the use of GM-1 led to an air content of 3% (43% greater than that of the control mortar). When the major components used in GM-1 were added separately, it was observed that only the yeast extract led to an increase in air entrainment, which is an 81% increase. This is most likely due to a low surface tension caused by the amphiphilic ingredients in the yeast extract [29].

In contrast, it was observed that the air content of fresh mortar containing either calcium acetate or sodium citrate was lower than that of the control mortar. The results from the surface tension tests are shown in Section 3.1. The reduction of the air content with the addition of calcium acetate or sodium citrate may be because they led to a solution with higher surface tension due to their strong ionicity than the control. Consequently, it can be deduced that increases in air content due to GM-1 were related to the use of yeast extract.

### 3.3. Fluidity of Mortar

The effects of the GM-1 on the fluidity are shown in Figure 2. Results were obtained from averages of triplicate samples. In general, the fresh mortar containing GM-1 had a slightly greater fluidity than the control mortar. However, this does not appear to have been as a result of the addition of any of the three individual components, which, when tested individually, had no significant effect on fluidity.

It was found that there was a positive correlation between the fluidity of the tested mortar and the setting time of the corresponding cement paste. However, if the setting time of two groups was similar, then the fluidity was influenced by the air content, most likely through a similar effect as that of air-entraining agents.

### 3.4. Capillary Absorption of Mortar

Capillary absorption of cement-based materials has a positive correlation with their permeability [30]. Continuous records of seven days capillary absorption of mortar samples are shown in Figure 3. Results were obtained from averages of triplicate samples. Except for sodium citrate, the addition of GM-1 or other components decreased the water absorption. Consequently, GM-1 would not increase mortar permeability. Although calcium acetate increased the surface tension from the control situation, there was also a lower permeability, as shown by the decreased capillary absorption. This indicates that the surface tension alone is not a good predictor of capillary absorption.

### 3.5. Strength Development of Mortar

The compressive strength of all mortars at 3 days and 28 days are shown in Figure 4. Results were obtained from averages of sextuplicate samples. The addition of GM-1 had little effect on the strength, compared to the control at either age. However, of the three major components, it can be seen that calcium acetate led to an improved compressive strength development. This may be because of the high solubility of calcium acetate, which brings an increase in the availability of calcium ions at an early age [31]. This characteristic is similar to many accelerators based on calcium chloride and calcium formate [32].

Both sodium citrate and yeast extract reduced compressive strength. In the case of sodium citrate, this is probably related to its effect on the setting. However, the retardation alleviated with time as strength differences between sodium citrate samples and the control were limited at both 3 days and 28 days. In the case of yeast extract, the lower strength may be related to the increased air content.

The flexural strength of all mortars at 3 days and 28 days are shown in Figure 5. Results were obtained from averages of sextuplicate samples. While the trends are less clear than those of compressive strength, the same general observations apply. Calcium acetate improves flexural strength while sodium citrate and yeast extract have a negative effect on flexural strength. In general, the effect shows that the control and the mortar containing GM-1 perform similarly at 28 days.

### 3.6. Setting of Cement Paste

Test results of the initial and final setting time are shown in Figure 6. Results were obtained from averages of triplicate samples. The control paste had an initial setting time of approximately 250 min and a final setting time of around 350 min. The addition of GM-1 led to a significant retardation of the setting – delaying initial set by a further 300 min and the final set by a further 400 min. Of the three major components, it can be seen that sodium citrate played a dominant role in retarding the setting of the cement. The retarding effect of sodium citrate was likely due to the adsorption of citrate ions on cement particles, in a similar manner to how citric acid affects the hydration of concrete [33].

Neither yeast extract nor calcium acetate had any significant effect on the setting of the cement at the proportions used.

### 3.7. Hydration Kinetics of Cement Paste

Figure 7 shows the rate of heat evolution against time for a control paste and pastes containing the addition of 0.1%, 0.3%, and 0.5% of GM-1. While the addition of 0.1% GM-1 led to a slight increase in the maximum rate of heat evolution, it may be generally concluded that the addition of GM-1 led to a retardation of the hydration, which is consistent with the delay in setting observed in Figure 6.

The effects of the three major individual components are shown in Figure 8. Both the calcium acetate and yeast extract appeared to accelerate the hydration and led to increased rates of heat evolution. However, sodium citrate retarded the hydration. Again, this was consistent with the results for the setting in Figure 6. On the other hand, the magnitude of retardation did not appear to be consistent with that observed in GM-1. Further tests were carried out with each of the minor individual components given in Table 1, but no effect on hydration could be found.

As a result of this, an alternative growth media, GM-2, was derived in which sodium citrate was eliminated (See Table 5). The results of isothermal conduction calorimetry using this growth media at 0.1%, 0.3%, and 0.5% of the mass of cement are shown in Figure 9. While there was still some delay in hydration when using GM-2 at 0.5% of the mass of cement, in general, the omission of sodium citrate from the growth media appears to eliminate most of the retardation effects.

It may be concluded that sodium citrate was the component of most single significance but that its use may also have led to a series of additional complex reactions resulting in far greater retardation than when used alone. As observed earlier, the retarding effect of sodium citrate was likely due to the adsorption of citrate ions on the cement particles by complexation with calcium ions, which prevents access to water [34,35,36].

### 3.8. Microstructure of Cement Paste

The microstructure of the neat cement paste and that containing GM-1 were observed at 7 days (Figure 10) and 28 days (Figure 11). Each figure shows the microstructure at a magnification of ×4000.

From the scanning pictures, the similarity in compressive strength of the neat cement paste and the cement paste with 0.5% GM-1 was further confirmed since they had similar microtopographies, even though there were some unknown crystals separated from the C-S-H structure (7 days, ×4000; 28 days, ×4000). The chemical compositions of the unknown materials require further investigation, but they are likely to be precipitates of the soluble growth media formed during drying.

## 4. Conclusions

The impact of microbiological growth components on the properties of cement pastes and mortars were investigated. While microbiological growth components for self-healing are designed to be encapsulated and protected from early-age concrete processes, there is a reasonable concern that these capsules could rupture during mixing or placing and release potentially harmful ingredients into the mixture. This paper reported on the effects of a multi-constituent growth media for bacteria on the basic properties of cement-based materials instead of the self-healing efficiency in a structural service. Specifically, this paper reported on the effects of three major components, which are calcium acetate, yeast extract, and sodium citrate.

It was shown that the use of calcium acetate as the precursor does not affect the setting and hardening of the concrete at the quantities that will likely to be released from the capsules. This suggests that it is eminently suitable as a self-healing concrete precursor and can be a viable alternative to calcium lactate. Yeast extract was shown to increase the air content of mortars with a knock-on effect on the compressive strength. Sodium citrate was found to be unsuitable as a supplier of sodium and carbon for self-healing because it significantly slowed the hydration kinetics and retarded the setting of the cement, even though such effects disappeared as time went on.

It was shown that the release of 10% multi-constituent growth media into early-age concrete (at 0.5% of the mass of cement) has no significant effect on the fluidity, setting, strength development, capillary absorption, or microstructure of the concrete when sodium citrate is not used. Additionally, pore structure and microtopography were not significantly affected by the released growth media.

These results are important since they demonstrate that a multi-constituent growth media will not have significant effects on the properties of concrete, even in the case where 10% of it, which approximately equals 0.5% of the cement mass, is accidentally released into a concrete mix. However, care must always be taken and this work has shown that it is necessary to ensure that the major compounds of healing agents do not impede hydration.

## Figures and Tables

**Figure 1 materials-12-01303-f001:**
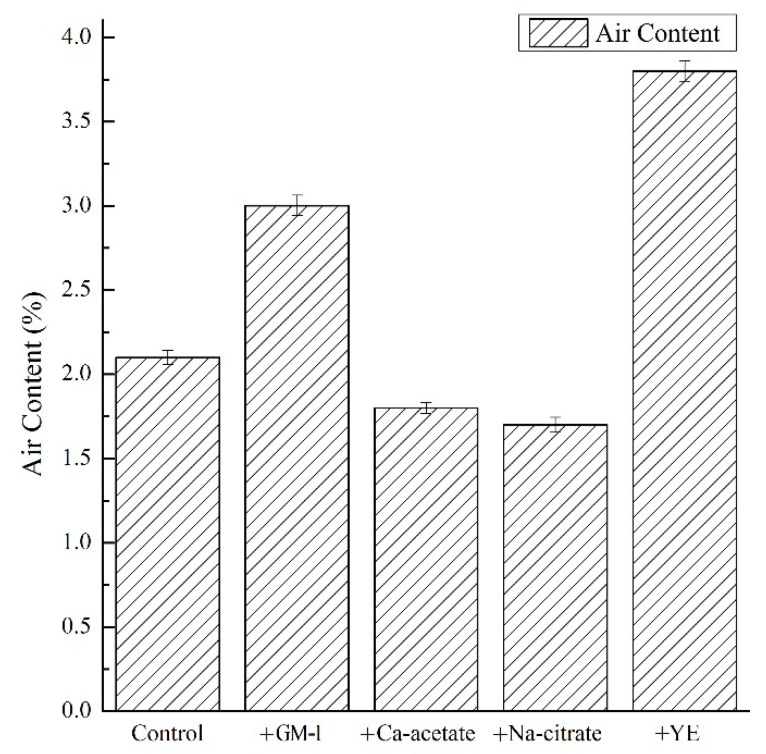
Air content of cement mortars.

**Figure 2 materials-12-01303-f002:**
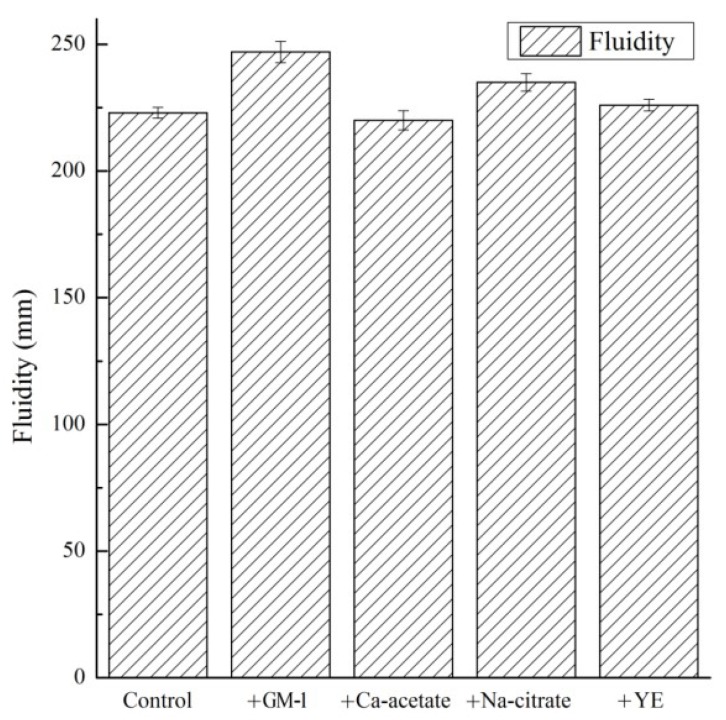
Fluidity of cement mortars.

**Figure 3 materials-12-01303-f003:**
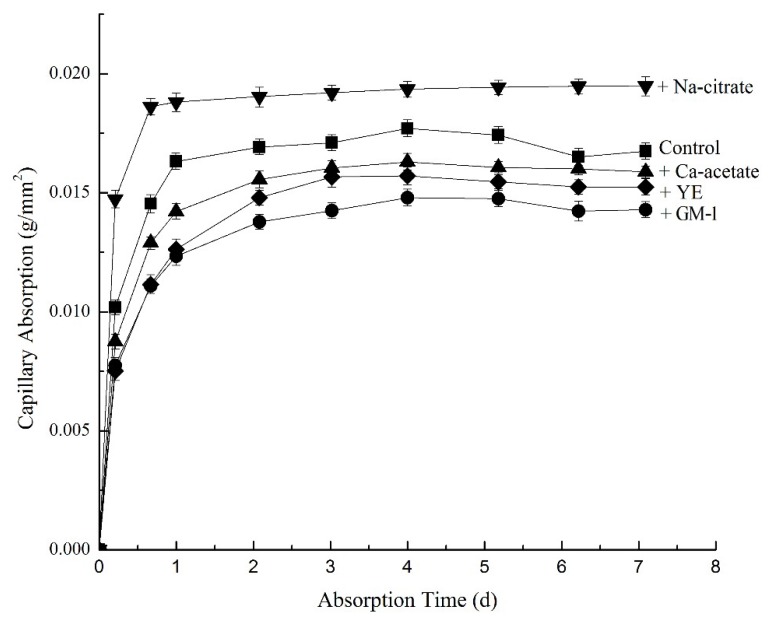
Capillary absorption of mortars.

**Figure 4 materials-12-01303-f004:**
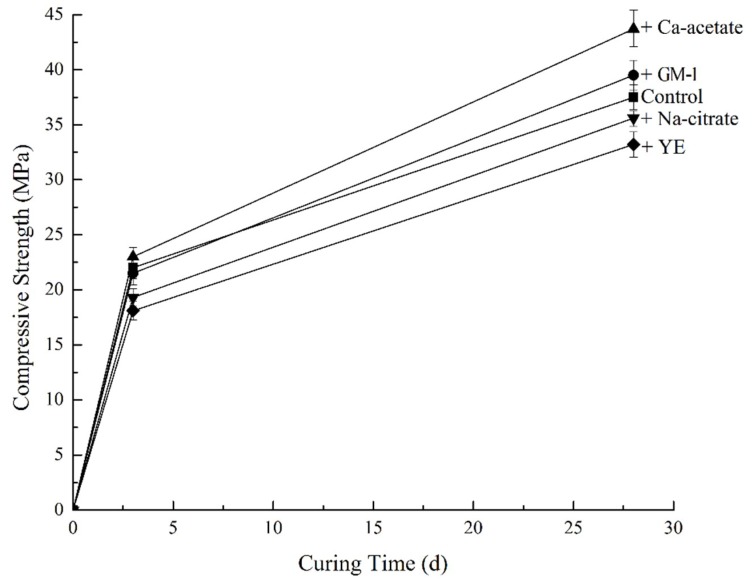
Compressive strength development of mortars.

**Figure 5 materials-12-01303-f005:**
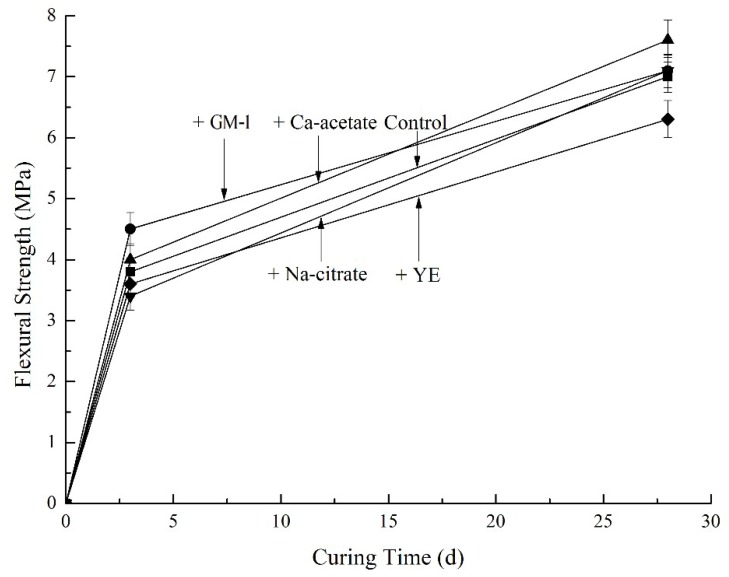
Flexural strength development of mortars.

**Figure 6 materials-12-01303-f006:**
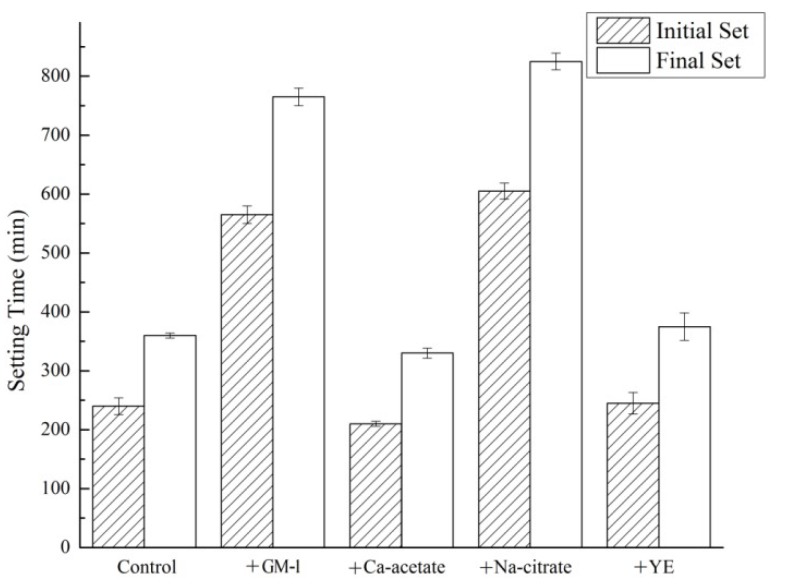
Initial and final settingtime of cement pastes.

**Figure 7 materials-12-01303-f007:**
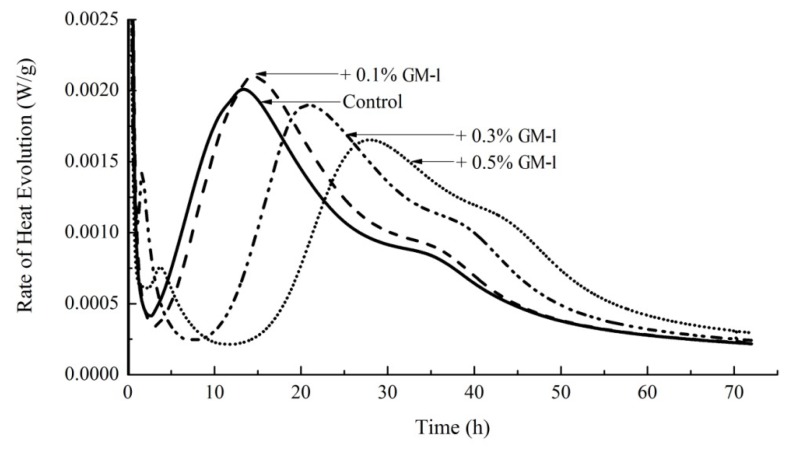
Hydration kinetics of cement paste containing GM-1at proportions of 0.1%, 0.3%, and 0.5% of the mass of cement.

**Figure 8 materials-12-01303-f008:**
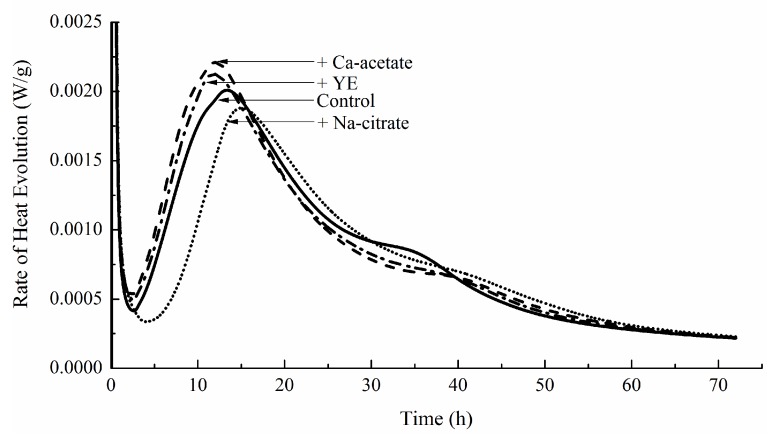
Hydration kinetics of cement paste containing Ca-acetate (0.28% of the mass of cement), yeast extract (0.06% of the mass of cement), and Na-citrate (0.09% of the mass of cement).

**Figure 9 materials-12-01303-f009:**
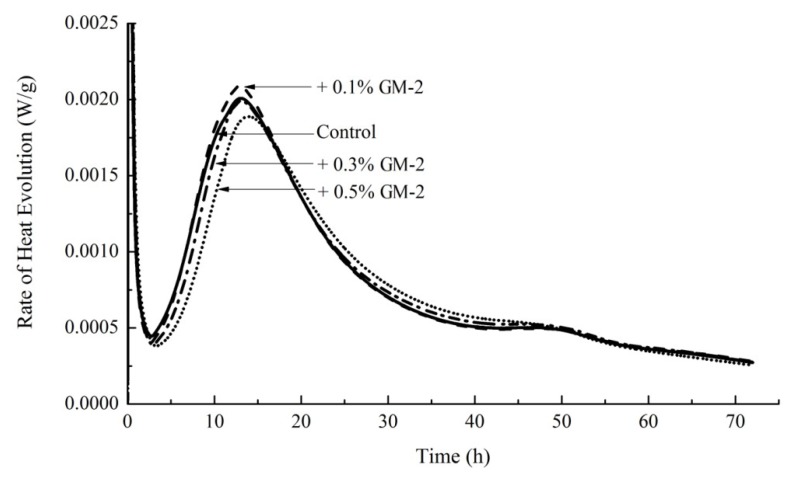
Hydration kinetics of cement paste containing GM-2 at proportions of 0.1%, 0.3%, and 0.5% of the mass of cement.

**Figure 10 materials-12-01303-f010:**
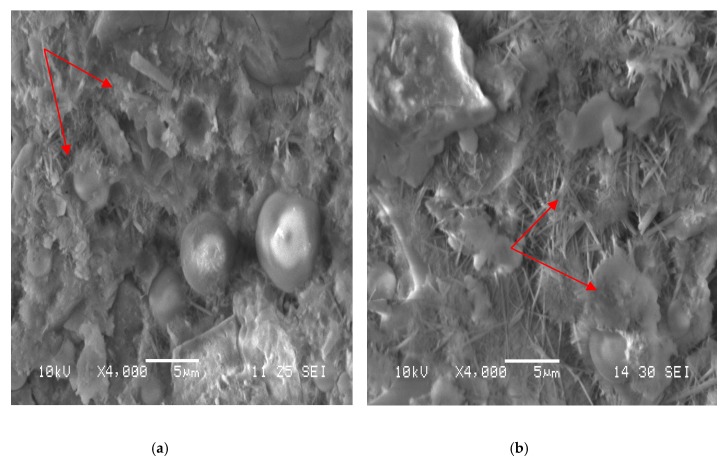
Microtopographyof cement paste at 7 days. (**a**) Neat cement paste. (**b**) Cement paste withGM-1 at 0.5% of the mass of cement.

**Figure 11 materials-12-01303-f011:**
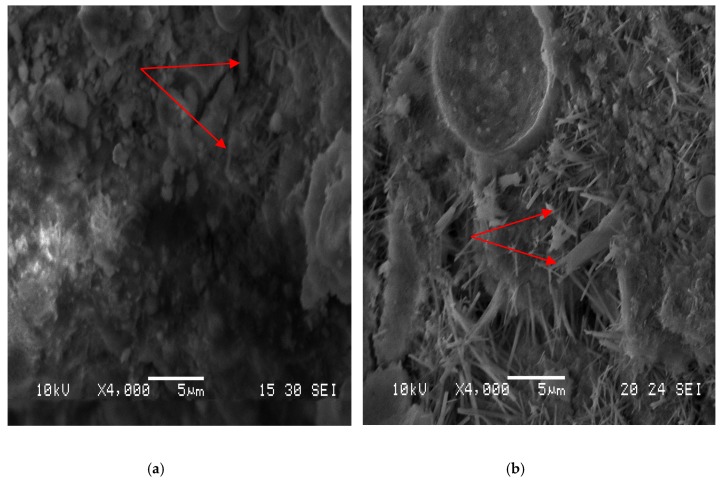
Microtopography of cement paste at 28 days. (**a**) Neat cement paste. (**b**) Cement paste with GM-1 at 0.5% of the mass of cement.

**Table 1 materials-12-01303-t001:** Functions of individual components of growth media.

Component	Function
Major Components	
Calcium acetate	Soluble calcium salt providingCa^2+^ for CaCO_3_ precipitation.
Sodium citrate	A source of carbon and energy.
Yeast extract	A complex source of nitrogen, vitamins, amino acids, and carbon (trace) for bacteria to grow.
Minor Components	
Sodium glutamate	An amino acid involved in protein synthesis and other fundamental processes such as glycolysis, gluconeogenesis, and the citric acid cycle. It is also a key metabolite linking nitrogen and carbon metabolism.
Alanine	Germination triggers, specific to Bacillus, which aid the formation of vegetative cells from spores.
Inosine	
Magnesium chloride	Used to activate enzymes and aids metabolism of carbohydrates, fats, and amino acids.
Sodium chloride	Controls the water activity that is required for the growth of vegetative cells and endospores.
Manganese sulphate	Mn is a key component required for sporulation (for repetitive self-healing).
Monopotassium phosphate	pH buffering agent.

**Table 2 materials-12-01303-t002:** Composition of tested growth media (% by mass).

Self-Healing Component	GM-1	Ca-Acetate	Na-Citrate	YE
Calcium acetate (Ca-acetate)	55.9	100.0	–	–
Sodium citrate (Na-citrate)	18.7	–	100.0	–
Yeast extract (YE)	12.1	–	–	100.0
Sodium glutamate	3.7	–	–	–
Alanine	1.9	–	–	–
Inosine	1.9	–	–	–
Magnesium chloride	1.9	–	–	–
Sodium chloride	1.9	–	–	–
Manganese sulfate monohydrate	1.2	–	–	–
Potassium hypophosphite	0.9	–	–	–
Total	100.0	100.0	100.0	100.0

**Table 3 materials-12-01303-t003:** Mix proportions for each mortar.

Group	Component/Cement Mass Ratio	Growth Media, g	Cement, g	Water, g	Standard Sand, g
Control	0%	0	450	225	1350
GM-1	0.50%	2.25	450	225	1350
Ca-acetate	0.28%	1.30	450	225	1350
Na-citrate	0.09%	0.43	450	225	1350
YE	0.06%	0.28	450	225	1350

**Table 4 materials-12-01303-t004:** Effects of growth media on surface tension of saturated solutions.

Group	Control	GM-1	Ca-Acetate	Na-Citrate	YE
Solute/Solvent ^1^	0/100	16.5/100	35/100	10/100	2/100
Density (g/cm^3^)	1.000	1.165	1.350	1.100	1.020
Height ^2^(mm)	145	102	118	140	120
Surface Tension (mN/m)	72.8	59.6	79.9	77.2	61.4

^1^ This is the saturated state for a solution; ^2^ This is the height of the liquid column (raised solution) in the glass capillaries in the tests.

**Table 5 materials-12-01303-t005:** Composition of GM-2 (% by mass).

Self-Healing Component	GM-2
Calcium acetate (Ca-acetate)	66.2
Sodium citrate (Na-citrate)	–
Yeast extract (YE)	20.5
Sodium glutamate	3.7
Alanine	1.9
Inosine	1.9
Magnesium chloride	1.9
Sodium chloride	1.9
Manganese sulfate monohydrate	1.2
Potassium hypophosphite	0.9
Total	100.0

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
