# Peer review of "Effect of Microbiological Growth Components for Bacteria-Based Self-Healing on the Properties of Cement Mortar"

_materials, 2019, doi:10.3390/ma12081303_

Round 1

Reviewer 1 Report

Please, find comments and suggestions in the attached file. 

Author Response

1.    The introduction section has been extended as suggested and some of the most recent advancements concerning the encapsulation methods suitable for self-healing concrete have been included in this section.

2.    References to GM-2 have been removed from Table 2 and there is only a quick mention to it in Section 2.2. All details on GM-2 have only been provided in Section 3.6.

3.    Error bars have been added in figs. 1, 3, 4, 5.

4.    Explanation of symbols in eq. 3 has been added.

5.    The revised manuscript has undergone English language editing by MDPI. All the errors pointed out by the expert have been corrected.

Reviewer 2 Report

This paper describes a study looking into how effective non encapsulated  microbiological growth components are when they are mixed into a mortar.

Some comments:

1) Why were the tests done on mortar rather then concrete? How would the presence of coarse aggregate potentially affect your results?

2) Growth media GM-2 is mentioned on Page 3 and in Table 2, but then ignored and the reason the media was not explored futher on is not explained again until page 9. The authors may consider revising the flow of the paper, as it was confusing to understand.

3) Was only one particular sample of mortar for each mix prepared (Table 3)? Is this sufficient?

4) A grammar revision is recommended.

5) The purpose of the separation is to prevent the hydration process of concrete to also hydrate the spores and nutrients. The purpose is to only have the capsules “rupture” with the presence of water AFTER the concrete has set. If the spores and nutrients are hydrated immediately during the mixing of concrete, before setting, and before any cracks are formed, the calcium carbonate formed only adds to the comprehensive strength of the concrete, similar to what components containing silica would do. It in fact has no effect to concrete at all in the context of “self healing” because there is nothing to heal when the spores and nutrients are no longer present as they were used up during the “hydration process”. If there is no effect on self-healing, why would someone add bacteria in this manner to a mix?

6) The primary purpose of adding bacteria to concrete/mortar is to promote precipitation of calcite in cracks. While the study has looked at the effects of adding the constituents in non-encapsulated form, the biggest flaw is the lack of discussion on how effective the non encapsulated ingredients actually are in healing cracks, or even how the results of this study (ie no significant effect on fluidity, setting, strength development, etc) could potentially influence the effectiveness of the components on the ultimate goal of healing cracks. Until this has been addressed, it is difficult to understand the relevance of this study, and I do not recommend it go forward for publishing until this has been made clear. 

Author Response

We think there is a misunderstanding about aims and objectives of the manuscript in the review from the expert. The study is NOT aiming to investigate the healing efficiency of non-encapsulated microbiological growth components. Actually, it investigates the effects of accidentally released bacteria growth media from capsules in fresh cementitious mortars on the properties of the fresh/hardened composite. (Due to shearing action in the concrete mixer, there is a chance that capsules/carriers of self-healing agents may accidentally rupture and release their cargo.) Although we direct added growth components in fresh mortar or cement paste, the growth components were not used to build up a direct-mixed self-healing system. They were used to simulate accidentally released cargo from capsules during mixing.

1) The tests were done on mortars because the potentially negative effects of accidentally dispersion of bacteria growth media are mainly on the cement hydration. We considered mortar tests are more suitable than concrete tests for researches of cement hydration because the presence of coarse aggregates may increase the discreteness of the results. Also, the released components would not react with aggregates during hydration. (The study is not aiming to investigate a kind of self-healing concrete, in which the self-healing agents are mixed directly without any encapsulation.)

2) The flow of the paper is revised as suggested.

3) More replication experiments have been done and the updated results are obtained from averages of triplicate samples.

4) A grammar revision has processed. The revised manuscript has undergone English language editing by MDPI.

5) We totally agree with what the expert say and we have claimed that it is good to encapsulate and separate the self-healing ingredients. Components that accidentally released before setting have no significant effect on self-healing, therefore we never investigate the self-healing efficiency in the manuscript. Even though the components are encapsulated, there is still a chance that capsules/carriers may rupture and release their cargo due to the shearing action in the concrete mixer. Therefore, what we investigate is whether or how these ACCIDENTALLY released components detrimentally affect BASIC PROPERTIES of mortars.

6) From Comment 5 from the expert and the corresponding response, neither the expert nor us think the accidentally released components would contribute significantly to cracks healing. The study is NOT looking at the healing efficiency of non-encapsulated form but at the potentially detrimental effects of accidentally released components on basic properties of mortars. 

Round 2

Reviewer 2 Report

Lines 104-105 in the manuscript state:

Consequently, the aim of this research was to investigate the effects of growth components for bacteria-based self-healing (both individually and as a multi-component
addition)

This is where the confusion arises. While the authors have clarified the confusion in your response to the reviewer, I feel that the reason for conducting the research needs to be more explicitly mentioned in the paper (perhaps in the abstract, introduction and/or conclusion, as the authors best see fit), and that as the authors mention in their response, "The study is NOT looking at the healing efficiency of non-encapsulated form but at the potentially detrimental effects of accidentally released components on basic properties of mortars".

Author Response

The abstract, the introduction and the conclusion have been modified in order to clarify the confusion as what the reviewer suggested. The aim of the study has been more explicitly mentioned in the paper. Modified sentences are highlighted with yellow.